# Disordered Eating among People with Schizophrenia Spectrum Disorders: A Systematic Review

**DOI:** 10.3390/nu13113820

**Published:** 2021-10-27

**Authors:** Anoop Sankaranarayanan, Karthika Johnson, Sanop J. Mammen, Helen E. Wilding, Deepali Vasani, Vijaya Murali, Deborah Mitchison, David J. Castle, Phillipa Hay

**Affiliations:** 1Western Sydney LHD Mental Health Service, Blacktown, NSW 2148, Australia; Karthika.Johnson@health.nsw.gov.au (K.J.); Sanop.Mammen@health.nsw.gov.au (S.J.M.); Deepali.Vasani@health.nsw.gov.au (D.V.); Vijaya.Murali@health.nsw.gov.au (V.M.); 2Translational Health Research Institute, School of Medicine, Western Sydney University, Campbelltown, NSW 2560, Australia; deborah.mitchison@westernsydney.edu.au (D.M.); P.Hay@westernsydney.edu.au (P.H.); 3St. Vincent’s Hospital Melbourne, Fitzroy, VIC 3065, Australia; Helen.Wilding@svha.org.au; 4Centre for Addiction and Mental Health, University of Toronto, Toronto, ON M5S 2E8, Canada; David.Castle@camh.ca

**Keywords:** disordered eating, binge eating, night eating, food craving, food addiction, schizophrenia, schizoaffective disorder

## Abstract

Disordered eating, or abnormal eating behaviours that do not meet the criteria for an independent eating disorder, have been reported among people with schizophrenia. We aimed to systemati-cally review literature on disordered eating among people with schizophrenia spectrum disorder (SSD). Seven databases were systematically searched for studies that described the prevalence and correlates of disordered eating among patients with SSD from January 1984 to 15 February 2021. Qualitative analysis was performed using the National Institutes of Health scales. Of 5504 records identified, 31 studies involving 471,159 subjects were included in the systematic review. The ma-jority of studies (17) rated fair on qualitative analysis and included more men, and participants in their 30s and 40s, on antipsychotics. The commonest limitations include lack of sample size or power calculations, poor sample description, not using valid tools, or not adjusting for con-founders. The reported rates were 4.4% to 45% for binge eating, 16.1% to 64%, for food craving, 27% to 60.6% for food addiction, and 4% to 30% for night eating. Positive associations were re-ported for binge eating with antipsychotic use and female gender, between food craving and weight gain, between food addiction and increased dietary intake, and between disordered eating and female gender, mood and psychotic symptoms. Reported rates for disordered eating among people with SSD are higher than those in the general population. We will discuss the clinical, treatment and research implications of our findings.

## 1. Introduction

People with schizophrenia have a two to five-fold increased risk for diabetes, metabolic syndrome, obesity, and cardiovascular diseases compared to the general population [1,2]. Up to 25% of people with schizophrenia have diabetes mellitus, 35% have metabolic syndrome, and up to 50% of those on antipsychotics gain weight [3,4]. These adverse physical health outcomes contribute to a 20% (20–30 year) reduction in lifespan [5] mainly through cardiovascular sequelae [6,7]. These physical health issues are also associated with other negative outcomes such as poor adherence to treatment [8,9], poor quality of life [10,11], greater risk of relapse [12,13], and low self-esteem [11].

Extant research has shown consistent associations between diabetes, obesity, and metabolic syndrome, on the one hand, and behaviors such as smoking, substance use, unhealthy dietary choices, and a sedentary lifestyle, on the other [7,14]. People with schizophrenia are more likely to have a diet rich in saturated fats and calories and poor in fibre and fruit [1]. It is also well established that rates of cigarette smoking (60–90%) and substance use [15] are much higher amongst people with schizophrenia, than the general population. Impaired executive functioning [16] has also been found to be associated with higher rates of metabolic adversities.

A relevant and common problem is disordered eating, which is a term used to describe unhealthy eating patterns that may not meet the diagnostic threshold for an eating disorder, but which have a negative impact on functioning [17]. Studies in non-psychotic and general populations provide evidence for associations between disordered eating and childhood adversities [18], substance use disorders [19], type 2 diabetes mellitus [20], obesity [21], executive function deficits [22] and frontal lobe functional impairment [23].

As all these adversities are far more common among people with schizophrenia spectrum disorders, in conjunction with the greater likelihood of people with schizophrenia to have a poor diet as outlined above, it is conceivable that rates of disordered eating behaviors are higher in this population. Interestingly, over 100 years ago, Kraepelin and Bleuler described disorganized and uncontrolled food intake as one of the clinical characteristics of schizophrenia [24,25]. Bleuler associated this with delusions of poisoning, agitation, autism, and catatonic negativism [26]. The topic has found a renewed interest recently, and several studies have now examined the prevalence and correlates of disordered eating in schizophrenia. Studies have described associations between eating behaviors and medications such as clozapine and olanzapine [27], as well as with executive dysfunction [28], positive symptoms [29], and anxiety, depression, and sleep disturbances in people with schizophrenia [17]. Other contributors to disordered eating in schizophrenia include abnormalities of the HPA axis [26] and shared genetic vulnerability for eating disorders and schizophrenia [30]. Some authors have posited disordered eating as being an independent and distinct entity amongst people with schizophrenia [29].

Published studies of eating behaviors in schizophrenia differ widely in terms of their methodology and aims, the type of disordered eating studied, the instruments used, and the sample selected. It is therefore important to review these studies systematically to understand the association and mechanisms, with a view of guiding clinical practice and future research. In regard to previous reviews related to this topic, Dipasquale et al. [1] systematically reviewed the literature on dietary patterns in schizophrenia, rather than disordered eating per se. Their review of 31 studies found that schizophrenia patients’ diet was high in saturated fat, and low in fibre, factors that contribute to a higher risk of metabolic abnormalities. Kouidrat et al. [31] reviewed the literature on eating disorders (rather than disordered eating) in schizophrenia in 2014. They concluded that binge eating disorders and night eating syndromes are frequently found in people with schizophrenia and recommended regular screening for eating disorders in schizophrenia patients, but that review is now dated. Finally, and more recently, Stogios et al. [32] performed a scoping review of disturbed eating in psychosis. While this is current, the study had a broad scope of including all psychotic disorders (including bipolar disorders) and all eating behaviors (including food preferences and dietary compositions) without assessing the strengths or weaknesses of individual studies, or providing a qualitative analysis of the reviewed studies. The authors focused on the psychopathological and neurobiological mechanisms as opposed to uncovering the phenomenology of eating disturbances in psychotic disorders.

Hence, there is a need for an updated, systematic review of disordered eating among people with schizophrenia spectrum disorders (SSD). We present such a review and report specifically on the prevalence and correlates of binge eating disorder (diagnosed, sub-threshold, and spectrum disorders), food craving and food addiction, night eating, and other disordered eating behaviors including loss of control over eating, as well as cognitive restraint. In particular, we determine the prevalence of aberrant eating behaviors and assess whether they are related to the specific symptoms/domains of schizophrenia symptomatology (e.g. positive, negative, or cognitive or affective) or stage of illness (acute versus chronic). We also assess associations with specific antipsychotic treatments. This information we anticipated will help clinicians identify and manage disordered eating in people with SSD.

## 2. Materials and Methods

Our systematic review followed the Preferred Reporting Items for Systematic Reviews and Meta-Analysis (PRISMA) [33] guidelines. Publications were identified through a systematic search of seven bibliographic databases: Ovid MEDLINE(R) ALL 1946 to 15 February 2021; Embase 1974 to 2021 February 15 (Ovid); APA PsycInfo 1806 to February Week 2 2021 (Ovid); Ovid Emcare 1995 to 2021 Week 05; CINAHL (EBSCOhost); Cochrane Library (Wiley) and the Informit Health Collection and Humanities and Social Sciences Collection (RMIT).

Search strategies were developed by a medical librarian (HW) in consultation with AS, DC, DM and PH. Strategies combined the concepts of Schizophrenia AND Disordered Eating using a combination of subject headings and text words. Searches were limited to English language publications dated 1 January 1980 onwards: we chose this cut-off to coincide with the publication of DSM III, as there were changes to the nosology of the eating disorders, notably bulimia nervosa being added as a discrete disorder. An initial search was developed for Ovid Medline (Table 1) and then adapted for other databases, adjusting subject headings and syntax as appropriate. All bibliographic database searches were updated on 17 February 2021 except for Informit Health, which ceased to exist in the same format and was last searched on 27 August 2020. Reference lists of included studies were screened for additional publications.

### 2.1. Study Selection

Search results were exported to Endnote bibliographic management software and duplicates removed. In accordance with pre-determined inclusion and exclusion criteria (Table 2), records were screened on publication type. Conference abstracts, conference papers, dissertations, irrelevant letters, notes, comments, and book reviews were excluded. All remaining records were uploaded into Covidence systematic review software for further screening. Two reviewers (AS and DV) independently screened the remaining records on title and abstract, then full text based on the predetermined inclusion and exclusion criteria (Table 2). Conflicts were resolved through discussion between both reviewers and, where necessary, a third reviewer.

### 2.2. Quality Appraisal

AS and SM undertook the quality appraisal, using items from the National Institutes of Health (NIH) Study Quality Assessment Tools for Case-Control studies, and NIH Study Quality Assessment Tools for Observational Cohort and Cross-Sectional studies [34]. The following items were focused on: (1) type of study design; (2) clear research question with clear aims, objectives, or hypothesis; (3) sample size justification or power calculation; (4) clearly defined study population (inclusion and exclusion criteria and details of the study subjects and setting were discussed); (5) clearly defined and valid outcome measures (objective measures and instruments that are reliable and valid); (6) analyses use appropriate statistical techniques and account for all participants; (7) key confounders measured and adjusted for. Some of the items on NIH scales (e.g., exposure measured prior to the outcome; whether the timeframe was sufficient to expect an association between exposure and outcome; were different levels of exposure measured; and blinding) were less relevant to this systematic review and were therefore dropped. The chosen items allowed for a maximum of 9 points and studies were ranked as poor (1–3 points), moderate or fair (4–6 points), or good in quality (7–9 points) depending on the total points scored.

### 2.3. Data Extraction

A data-extraction checklist was created and AS, KJ, and VM extracted data on: study details (first author’s name, year of publication, country studied in), study design, sample characteristics (age and gender distribution, sample diagnosis and diagnostic criteria studied, selection criteria, study setting; and, for case-control studies, the description of controls), outcome variables studied (definition, instruments used, diagnostic criteria if relevant), any confounders adjusted for, and information on missing variables. All papers were reviewed by at least two investigators.

### 2.4. Data Synthesis

Data were synthesized descriptively.

## 3. Results

Figure 1 summarized the search and selection process. Our final selection included 31 studies [17,26,28,29,30,35,36,37,38,39,40,41,42,43,44,45,46,47,48,49,50,51,52,53,54,55,56,57,58,59,60] (Table 3). Taken together, these studies included 471,158 individuals (including a single study by Striegel-Moore et al. [58] that encompassed 466,590 men), of whom 2651 had a diagnosis of SSD. A number of studies included patients with SSD as well as other psychiatric disorders but did not describe the exact break-up of the included diagnostic categories [26,28,43,57,58], making it impossible to determine the exact numbers of patients with SSD.

The majority of the studies were cross-sectional observational studies (*n* = 15) [30,36,37,38,42,43,45,49,50,52,53,55,56,58,59] followed by case-control studies (*n* = 9) [17,26,29,35,39,44,47,48,51]. The case-control studies recruited healthy controls without history of mental illness. There were four cohort studies [40,41,54,60], with follow-up periods ranging from 10 weeks to 6 months. Three studies [28,46,57] reported results from clinical trials. Most of the studies were from Europe (47%), followed by North America (23%), although there were studies from most regions. Two studies were performed across multiple sites [28,60].

### 3.1. Sample Characteristics

Twenty-seven studies [17,26,28,29,30,35,36,37,38,39,40,41,42,44,55,57,60] were performed in patients with a diagnosis of schizophrenia or SSD, and four studies included patients with multiple diagnoses [43,56,58,59]. The majority of the studies used DSM-IV criteria (*n* = 20), followed by ICD-10 (*n* = 3); one multisite study [60] used both DSM-IV and ICD-10 criteria. Five studies did not describe the diagnostic criteria used [28,38,42,43,56]. In around half of the studies (*n* = 13), participants were recruited from community or outpatient settings [29,36,38,42,44,45,48,50,51,52,53,56,60], while eight studies employed hospital or inpatient settings [17,26,30,40,43,49,58,59], and three studies recruited from multiple locations [35,41,55]. Seven studies [28,37,39,46,47,54,57] either were unclear or did not specify where participants were recruited from.

The majority of the studies (*n* = 15) included more men [28,29,30,35,37,38,39,41,42,47,48,50,53,55,59]. All studies were of adults, apart from the study of Bachman et al. [37], which was of a child and adolescent population with a mean (SD) age of 19.2 (2.3) years for males and 19.7 (2.6) years for females. The majority of the other studies included participants who were in their 30s [30,35,39,40,41,44,45,46,47,49,53,54,55,60] and 40s [26,36,38,48,51,52,56,57]. Case et al. [28] described data from four trials and participant age varied across sites. Goluza et al. [42] and Kurpad et al. [50] did not provide the mean age for their sample and two studies [43,58] included a mix-sample with no specific age descriptions for participants with SSD.

The majority of the participants in the included studies were receiving antipsychotics, mostly second-generation antipsychotics and specifically, olanzapine and clozapine. Only two studies recruited antipsychotic-naïve patients [29,55]. Participants in most studies were overweight or obese [28,29,36,39,41,49,50,51,52,53,57]. Sixteen studies described limited or no selection criteria [26,28,30,37,38,40,42,43,44,46,49,51,53,56,57,58]. The commonly described exclusion criteria included those with substance use [36,45,48,49,50,51,55], diabetes mellitus [29,36,47,48,55], intellectual disability [17,29,47,48,59], and pregnancy [29,36,49,55].

### 3.2. Quality Appraisal

The results of the quality appraisal are summarized in Table 4. The majority (*n* = 17) of the articles were rated fair or moderate in quality [26,30,37,38,39,42,44,46,47,48,49,50,51,52,55,56,57]. Eight rated good [17,29,35,41,45,54,59,60] and six were poor [28,36,40,43,53,58]. There appeared to be a “period effect” with all of the studies rated “good” having been published after 2009, when we arbitrarily used three publication periods:1980–1999, 2000–2009, and 2010 and after. The type of reporting (e.g., short report by Striegel-Moore et al. [58]) and the study design (e.g., Kluge et al. [46] or Beaurepaire [38]) had an impact on the study ratings. Treuer et al. [60] was the best reported and Case et al. [28] rated poor. Although this study had a sample size of 805 subjects across the four trials, there were differences in design, duration of trial, sample characteristics, the dose of olanzapine used, and the tools used across the four trials that affected the quality of the study.

The majority of the studies present neither sample size nor power calculations. Other limitations include inadequate sample description [26,28,30,37,38,40,42,43,44,46,49,51,52,53,56,57,58], not defining outcome variables or not using valid tools [28,36,38,40,46,50,57,60]. Only 15 studies adjusted for confounders [17,29,30,35,39,41,47,48,52,54,55,56,57,59,60], most commonly obesity/BMI [39,41,48,52,55,56,57,59,60], age [41,48,52,54,56,57,59] and gender [41,48,52,54,56,57,59].

### 3.3. Disordered Eating

Most studies that reported or described disordered eating used validated and standardized instruments such as the Eating Inventory or Three Factor Eating Questionnaire (TFEQ) [37,39,45,47,48,55], Eating Attitudes Test (EAT) [17,26,29,56,59], Food Craving Inventory (FCI) [28,35,41], Dutch Eating Behavior Questionnaire (DEBQ) [55]. Others used diagnostic criteria for eating disorders from the DSM [26,38,43,44,45,53] or ICD [58]. Some studies reported using specially developed questionnaires [36,38,46,47], or tools that have not been validated, such as the Eating Behavior Assessment and Eating Attitude Scale [28,57].

#### 3.3.1. Binge Eating

Eleven studies [26,28,36,38,40,44,45,46,50,53,60] reported on binge eating. All of these studies reported patients who were on antipsychotics. Eight of these were cross-sectional, one case-control, and the other two reported secondary results from multi-site clinical trials. In summary, these studies indicate that people with SSDs on antipsychotic treatment had elevated rates of binge eating, with rates ranging from 4.4% [38] to 16% [40,53] for binge eating disorder (BED) and 8.9% [46] to 45% [50] for binge eating symptoms (not meeting the diagnostic threshold for BED). In general, cross-sectional [45,50,53], case-control studies [26,44] and studies with small sample size [40] reported higher prevalence rates. Aguiar-Bloemer et al. [36] and Bromel et al. [40] describe an increase in binge eating after the commencement of clozapine (through qualitative survey or prospective measures). One patient in the study of Bromel et al. described that their binge eating symptoms completely reverted after temporarily stopping clozapine but recommenced with re-initiation. In these studies, binge eating overall was associated with female gender [26,38,45] or weight gain [28,60].

#### 3.3.2. Food Cravings and FOOD Addiction

Eleven studies [28,35,36,40,41,42,46,49,50,54,57] provided information on food crav-ings and/or food addiction. All these studies were in people prescribed antipsychotic medication; four [36,42,49,50] were cross-sectional and one case-control [35]. The re-maining six used prospective data or were post hoc analyses of patients recruited to clinical trials. Prevalence rates for food cravings varied from 16.1% [36] to 64% [46] and was highest for sweets (chocolates, candies). Food addiction was reported in 27% [42] to 60.6% of samples [49]; Goluza et al. [47] noted that 77.4% of their patients’ reported symptoms of food addiction but did not experience associated distress or impairment. Study design could not account for the variation in rates seen. One study [35] found that patients on typical antipsychotics had higher craving scores than those on olanzapine (although not statistically significant). Craving for fast-food was associated with weight gain [28,41], and food addiction was associated with increased dietary in-take [49]. Stauffer et al. [57] demonstrated that reduction in carbohydrate craving was predictive of weight loss in treatment trials. Food craving inventory (28, 35, 41) and Yale food addiction scale (42, 49) were the most commonly used tools while some studies (36, 40, 50, 57) did not use validated tools.

#### 3.3.3. Night Eating

Night eating in SSD was assessed in four studies [38,50,51,52] two of which [51,52] were of overweight or obese individuals. The prevalence of night eating ranged from 4% [50] to 30% [38]. It is possible that the high rate reported in the study of de Beaurepaire [38] reflects their employing only a single screening question for night eating. However, Lundgren et al. [51] reported a rate of 27.2% for evening hyperphagia and 50% of their sample reported snacking on waking up in the middle of the night. Palmese et al. [52] reported that their patients were not aware of night eating but many endorsed symptoms of insomnia and depression. The four studies used different tools to study night eating.

#### 3.3.4. Other Disordered Eating Behaviors

A total of 18 reviewed studies [17,26,29,30,36,37,39,43,47,48,50,51,55,56,57,58,59,60] described various aspects of disordered eating behaviors (including snacking, impaired cognitive restraint, disinhibition, and susceptibility to hunger, scoring positive on scales for disordered eating such as Eating Attitudes Test (EAT) or Three Factor Eating Questionnaire (TFEQ), EDNOS, etc.) in people with SSDs. The reported prevalence ranged from 10% [30] to 41.5% [17] and was 2.4–4 times higher than in healthy controls [17,29] in case-control studies. There were methodological differences across studies; for example, while Fawzi and Fawzi [29] looked at current symptoms of DEBs, Malaspina et al. [30] assessed DEBs with onset prior to the SSD.

In general, people with SSD were more likely to score high on cognitive restraint, disinhibition, and hunger [37,39,47,48,55], to describe emotional eating [48,55], loss of control over eating [39,47], eating in response to internal and external stimuli, including sight, smell, or taste of food rather than hunger [39,51], and were likely to snack when unwell [36,50]. Many reported these issues were exacerbated by antipsychotics [17,37,55]. Other factors associated with disordered eating among people with SSD include female sex [26,29,30,55,56], anxiety and depression [17], prominent positive psychotic symptoms [17,30], negative symptoms [29], impaired executive function [47], and type 2 diabetes mellitus [17]. Elevated BMI was positively associated with disordered eating [29,37,55,56], albeit results were not uniform; for example, Fawzi and Fawzi [29] did not find any association between disordered eating and positive or negative symptoms, Kouidrat et al. [48] reported an association between higher cognitive restraint and BMI lower than 25, and Malaspina et al. [30] reported an association between DEBs and higher cognitive functioning, especially in men.

## 4. Discussion

We undertook a systematic review of disordered eating among people with SSDs. We found 31 studies encompassing 471,158 individuals, of whom 2651 had a reported diagnosis of an SSD. Two of these studies [43,58] included mixed diagnosis (including schizophrenia patients) among those with disordered eating.

Our review of studies, most of which were done in people receiving antipsychotics, found elevated rates of binge eating, night eating, food addiction, food craving, and disordered eating behaviors among people with SSD. Taken together, these studies report a prevalence of 4.4% to 16% for binge eating disorder (BED) and 8.9% to 45% for binge eating symptoms, 16.1% to 64% for food craving, 27% to 77.4% for symptoms of food addiction, 4% to 30% for night eating, and 10% to 41.5% for disordered eating behaviors. The rates for other DEBs were 2.5 to 4 times higher than that in healthy controls. Recent reviews [61,62] suggest lifetime prevalence rate for BED of 1.53% (95% CI 1–2.17) in the general population, and a point prevalence of 1.2–8% for BED and 3.8% to 8.4% for NES in people with type 2 diabetes mellitus. Pursey et al. [63] reported a weighted prevalence of nearly 20% for food addiction in obese adults. A simple comparison shows that the rates for disordered eating are higher in people with SSD; thus, there is a 2 times higher rate for BED, and up to threefold higher for night eating syndrome and food addiction among people with SSD. We also found that patients with SSD often do not fulfil the full diagnostic criteria for these conditions and in particular, do not report distress; hence they may have symptoms without the full diagnosis, and the rates for these are even higher. This lack of distress has been attributed to apathy and negative symptoms of schizophrenia, or the general lack of insight in this population [42], or because specific items such as “eating alone because of feeling embarrassed” did not apply to this population as they tended to live alone [38]. Previous studies in general population have shown that the incidence of DEBs is higher than diagnosable eating disorders, but despite their adverse impact on the individual’s functioning [64], are often untreated or missed by healthcare professionals [65]. There is also evidence for an association between disordered eating behaviors and excess eating, weight gain, and association with gender, and symptoms or chronicity.

It is unclear whether disordered eating in SSD is a part of the psychopathology, or whether it is secondary to medications, or an independent comorbid entity. We examine evidence for all three possibilities, below.

### 4.1. Role of Medications

Antipsychotic use, especially second-generation antipsychotics (SGA), is associated with higher risk of weight gain, obesity and metabolic complications [66]. A number of factors including neurotransmitter mechanisms, affect the brain’s satiety center, whilst an impact on insulin and gastrointestinal hormones and lifestyle issues have been put forth to explain this association [17]. In addition, disordered eating, such as binge eating and sweet food craving, have been reported in association with SGAs [27,46,67]. Studies that used a prospective design [40,46] demonstrated an increase in food cravings or binge eating over time for people on SGAs. Bromel et al. [40] reported a temporary cessation of binge eating in one patient when the antipsychotic was temporarily ceased. Using a case-control design, Blouin et al. [39], demonstrated a higher tendency for hunger and disinhibition scores for those on SGAs than healthy controls, and Sentissi et al. [55] reported increased reactivity to external food cues for people on SGAs compared to those on typical antipsychotics. These above factors increase the risk of excessive eating and loss of control over eating [44]. It is likely that antipsychotic-induced reduced serotonergic neurotransmission might also play a role in binge eating and food craving [46]; this includes evidence for reduced serotonin transporter binding among obese binge eating women in a SPECT study [68], a role of SSRIs in BED [69,70], and an association of serotonin transport gene polymorphism with BED [71]. Further, the effect of SGAs on the satiety center could account for why patients taking these agents continue to eat and do not feel full after a meal [51]. Schizophrenia patients with simultaneous high restriction and high disinhibition scores are more likely to be overweight and to be treated with atypical antipsychotics [44,55].

### 4.2. Illness Related

The fact that disordered eating behaviors were reported prior to the advent of antipsychotics [24,25,53] and in antipsychotic-naïve individuals [29] lends weight to the corollary that core psychopathology could play a role. This is further strengthened by findings that antipsychotic use did not impact binge eating patterns [50]. Historically, restrictive eating behaviors in schizophrenia were considered to be consequent upon delusions and hallucinations [26,51], whereas BED and BN were thought to be associated with negative, cognitive, and mood symptoms [50,53]. Indeed, disturbances in appetite, eating and weight are not confined to diagnoses classified under “eating and feeding disorders” in schemes such as the DSM (e.g., hyperphagia and deficient appetite noted in mood disorders). In their sample, Malaspina et al. [30] showed that patients who had a history of an ED had more severe psychotic and disorganization symptoms, whereas Treuer et al. [72] suggested that clinical improvement of psychotic symptoms seemed to coincide with increased food intake, while De Beaurepaire [38] reflect an association with improvement in psychotic symptoms. Indirect evidence suggests that perturbations in brain regions that moderate reward mechanisms and dopamine dysfunction in schizophrenia could play a role in food addiction [42,49]. This could also explain the sweet and fast-food craving seen in schizophrenia [73,74]. Alternate explanations for food cravings impairments in schizophrenia include dysregulation of serotonin neurotransmission [75] or altered dopamine/acetylcholine ratio in certain brain areas such as the nucleus accumbens [76].

Finally, the role of comorbid diabetes and disordered eating behaviors in schizophrenia needs to be considered and further studied. It is well accepted that rates of type 2 diabetes mellitus is 2–5 times higher in people with schizophrenia than in general population [77]. Although often linked to second generation antipsychotics, the first documented evidence for this association dates back to the late nineteenth century, long before antipsychotics and modern obesogenic diets [78].In a recent systematic review and meta-analysis exploring prevalence of diabetes in first episode psychosis population, Pillinger et al. [79] provide evidence for impaired glucose homeostasis from illness onset in schizophrenia. It is possible that diabetes and schizophrenia share common early developmental risk factors (e.g., preterm birth, gestational diabetes, maternal malnutrition, etc.) suggesting that stress and hypercortisolemia may contribute to the association between the two conditions.

Diabetes mellitus is associated with increased appetite and disordered eating (especially bulimia and binge eating disorder) and this, in turn, is associated with elevated HbA1C and other diabetic complications including retinopathy, neuropathy, lipid abnormalities, ketoacidosis and poor metabolic control [80]. Diabetes is also associated with impairment in satiety and hunger signals. Insulin crosses the blood-brain barrier and acts on hypothalamus to regulate appetite and food intake [81]. Further, anti-diabetic medications have been found to be effective in reducing binge-eating in non-diabetic population [82]. It is therefore possible that, at least in some individuals, antipsychotics (particularly second generation) triggers, increases, or unmasks the vulnerability for diabetes mellitus in people with schizophrenia and leads to changes in the appetite and weight control centers, predisposing the individual to disordered eating behaviors. The inherent symptoms of schizophrenia could further reinforce disordered eating. For example, negative symptoms and hypodopaminergia can induce food addiction, or binge eating, mood symptoms could be associated with comfort eating, and cognitive rigidity could be associated with reduced cognitive restraint.

### 4.3. Other Explanations

A third explanation for the association of schizophrenia with disordered eating, they are distinct entities that coexist and overlap by chance [29]. Disordered eating problems are often missed because clinicians focus on psychotic symptoms, and/or because the disordered eating is not severe or typical enough to meet the diagnostic threshold for an independent eating disorder label. Some commentators have conceptualized DEBs as a physiological compensatory mechanism in reducing psychotic symptoms; this is based on the premise that starvation induces psychosis [83]. DEBs often predate psychosis and share common risk factors including developmental trauma [38]. A shared genetic susceptibility for schizophrenia and anorexia nervosa has also been described [84].

### 4.4. Strengths and Limitations

The strengths of this review include the comprehensive search strategy and use of multiple authors for quality appraisal and data extraction. Limitations included limiting to only published literature (no grey literature) and papers in English. Whilst the impact of the latter may have been mitigated by the inclusion of multiple studies from non-English speaking backgrounds, adding grey literature can lower the impact of publication bias [85]. Our review also captured a different literature to the scoping review by Stogios et al. [32] and there were only ten studies in common [35,39,40,41,44,46,47,54,55,60]. We believe therefore, that our findings complement and lend additional weight to their findings.

We did not undertake a meta-analysis as the studies were highly heterogeneous, differing in the types of disordered eating studied, with differing methodologies, definitions, and thresholds. The quality of included studies was largely fair to good and we tried to link the information from individual data to the methodology, outcome measures, and qualitative analysis to give weight to the conclusions drawn. We had amended the NIH scale as some of the items were less relevant to our review and were therefore dropped. We do not believe this could have affected our quality rating.

While we did not exclude studies that focused on anorexia nervosa and bulimia nervosa, this review focused on disordered eating behaviors. There are overlaps between the different eating disorders and there is a possibility of diagnostic crossover or migration between the different disorders [86,87]. However, we are not aware of any studies that have necessarily studied the stability of eating disorder or disordered eating in SSD. Our review included studies from 1985 to 2021; our knowledge and understanding of eating disorders have evolved over this time and this is reflected in how the classificatory systems have changed. For example, the criteria for binge eating in Hay and Hall’s [43] study using DSM III R criteria was “at least two days a week for three months”, which changed to “at least two days a week for 6 months” in the DSM-IV, and now is at least one day a week for three months” in the DSM-5. Similarly, night eating syndrome, although initially described by Stunkard in 1955 [88], only found a place under OSFED criteria in the DSM-5. These changes have an impact on what is considered and accepted as disordered eating behaviors.

### 4.5. Clinical Implications

It is likely that multiple factors play a role in the association between schizophrenia with disordered eating. Antipsychotics influence the appetite and satiety centers of the brain and lead to hyperphagia [39,89]. There could be a simultaneous preference for sweet and hyperpalatable foods, which contain high dietary levels of sugar and fat. This can, in turn, have “high addictive potential” and reinforce further disordered eating behaviors [90]. Intrinsic illness factors in schizophrenia such as abnormalities in the mesolimbic reward circuit, or impaired cognitive restraint and executive functioning could mediate further disordered eating patterns in response to increased appetite and cravings.

It is possible that the actual prevalence of disordered eating in people with SSDs is substantially higher than reported in the studies reviewed here. Several factors could account for lower reported rates, including absence of distress or impairment [42], confusing criteria in people with schizophrenia (e.g., teasing out the differences between loss of control and intense desire, or between food craving and increased hunger), as well as lack of clear definitions (e.g., for food addiction). Further, de Beaurepaire [38] reported patients who experienced binge eating as pleasurable, in contrast with others who eat excessively but eat slowly. Hence it is likely that different thresholds or criteria are needed when assessing disordered eating behaviors in people with SSD. Similarly, future studies exploring night eating in schizophrenia should differentiate between nocturnal eating and evening hyperphagia.

Future studies need to examine the role of smoking in obesity-related behaviors [91,92]. SSDs are associated with high rates of cigarette smoking and substance use [15]. This has been partly explained by a putative self-medication hypothesis as a mechanism to offset the “negative” symptoms or hypodopaminergia. It is plausible that DEBs such as food addiction is a behavioral variant of this “impulse-control” issue and reflective of underlying impairments in the reward system. The possible role of smoking on cognition in schizophrenia and the impact on disordered eating needs to be studied further. Similarly, there is a need to study the role of smoking cessation or reduction on eating behaviors in people with SSDs. Studies have shown an association with all phases (acute and chronic) and all symptom domains of schizophrenia (positive, negative, cognitive, and affective). There is therefore a need for further research into disordered eating and in particular whether this forms an integral part of the symptoms of schizophrenia as discussed by Bleuler and Kraeplin more than a hundred years ago [24,25] and if disordered eating should be included among diagnostic criteria for schizophrenia. It is particularly important to look for “disordered eating” rather than an “eating disorder” as patients with SSD may not fulfil all the criteria for eating disorders, while they could still experience symptoms or manifest features of an eating disorder. An association between choking risk in schizophrenia (and in particular with antipsychotic use) [93] and choking risk with eating rapidly [94] have been described. The role of disordered eating behaviors in choking risk in SSD needs to be studied further.

As the rates for disordered eating are higher among people with SSD, it is important to train clinicians to look for, assess and manage eating disorders at various stages of the illness, particularly modelled along the lines of those for substance-use disorders [56]. Patients with SSD should be offered healthy lifestyle and dietary advice to develop skills to respond to physiological urges to eat, encouraged to eat on a regular schedule, eating on a budget, focus on reducing sugary food, and improve exercise and physical activity [51,56]. Cognitive behavioral therapy has shown to be effective for binge eating in a randomized controlled study of patients with antipsychotic-induced weight gain [95].

Pharmacological strategies have a role in mitigating antipsychotic induced weight gain. For example, preliminary research shows that naltrexone can be useful in antipsychotic induced weight gain [96] and targets both the endorphin and dopamine systems and could play a role in food cravings [97]. Naltrexone can reduce the rewarding aspects of food and reduce food consumption and reduce appetite [98]. It is important to consider that at least some studies have described that people with SSD did not report “distress” or “lack of impairment”; the role, place, and type of intervention needs to be studied further to elucidate whether intervention is warranted and what the best form of intervention is.

Studies have shown an association between SSD and higher scores on cognitive restraint, disinhibition, and hunger [37,39,47,48,55]; higher cognitive restraint is associated with lower BMI [48] and with weight loss [99] and better weight loss maintenance in treatment programs [100]. Studies of eating disorders have shown an association between cognitive flexibility and eating disorders, particularly anorexia nervosa [101,102] and a role for cognitive remediation therapy (CRT) for these disorders [103,104]. Although CRT has been studied in schizophrenia [105], its role in addressing disordered eating in schizophrenia has not, to our knowledge, been examined. Future research should explore the role for CRT in the different disordered eating in SSD.

## 5. Conclusions

Our systematic review provides evidence for an increased rate of binge eating, night eating, food cravings and addiction, and disordered eating behaviors among SSD patients. While this could be related to the illness itself, most of the included studies were performed in patients receiving antipsychotic treatment; hence, it would be difficult to establish whether this is a primary issue or secondary to medications or symptoms. There is a need for studies that assess disordered eating at baseline in early psychosis and prospectively with the commencement of antipsychotics. Studies need to be powered to be able to study the role of medications and other illness characteristics and should be able to inform the role of disordered eating in people developing metabolic complications.

## Figures and Tables

**Figure 1 nutrients-13-03820-f001:**
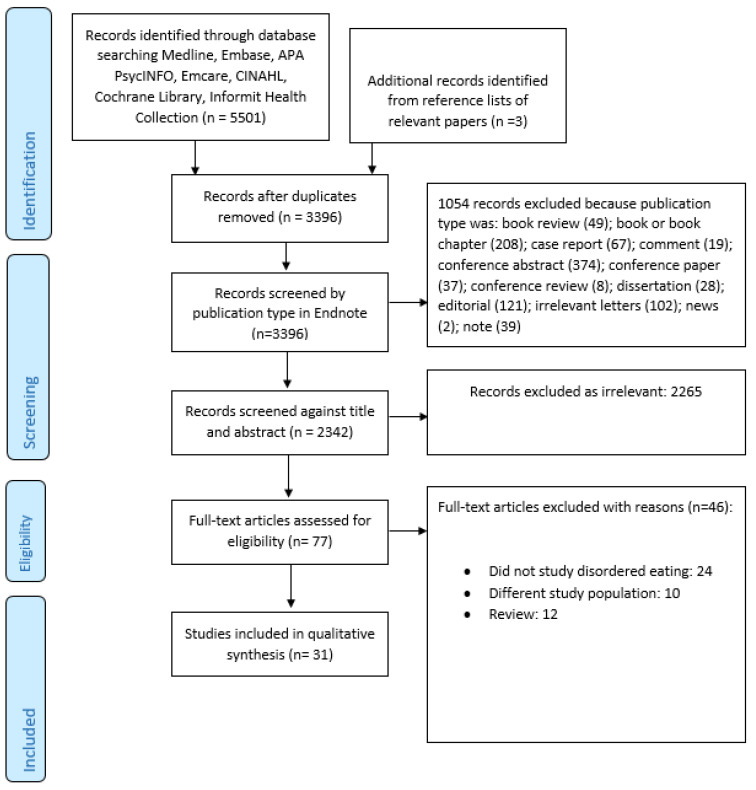
PRISMA flow diagram.

**Table 1 nutrients-13-03820-t001:** Search strategy for Ovid MEDLINE ^®^ ALL 1946 to 15 February 2021.

schizophrenia/ or schizophrenia, catatonic/ or schizophrenia, disorganised/ or schizophrenia, paranoid/
2. (schizophren* or schizoaffective or schizo-affective).ti,ab,kw,kf.
3. 1 or 2
4. “feeding and eating disorders”/or binge-eating disorder/or bulimia nervosa/or food addiction/or night eating syndrome/or bulimia/
5. (eating disorder* or disordered eating or eating disturbance* or eating behav* or over eat* or overeat* or compulsive eat* or excessive eat*).ti,ab,kw,kf. or eating.ti.
6. (bulimi* or binge eat* or binge purge or polyphagi* or food addict*).ti,ab,kw,kf.
7. (feeding disorder* or disordered feeding or night eating or nighttime eat* or nocturnal eat* or sleep related eating).ti,ab,kw,kf.
8. appetite/or appetite regulation/or *eating/
9. (food intake or appetite or unhealthy eating or unhealthy diet* or unhealthy food).ti,ab,kw,kf.
10. 4 or 5 or 6 or 7 or 8 or 9
11. 3 and 10
12. exp animals/not humans/
13. (animal* or rat or rats or mouse or mice or rodent* or monkey*).ti.
14. 12 or 13
15. 11 not 14
16. limit 15 to yr=“1980 -Current”
17. limit 16 to english language

**Table 2 nutrients-13-03820-t002:** Inclusion and exclusion criteria.

Inclusion Criteria	Exclusion Criteria
Participants: patients with a diagnosis of schizophrenia, schizoaffective disorder, schizophreniform disorders and first episode schizophrenia or includes Schizophrenia spectrum disorders. We also included studies that were done in people with mixed diagnosis provided they reported findings for people with SSD. Interest: We included studies that examined or described “loss of control over eating”, binge eating, disordered eating, night eating/nocturnal eating, obesity, overweight, bulimia nervosa, or disordered eating Comparison: to those without disordered eating or healthy controls. Outcome: We included studies that reported prevalence and correlates of disordered eating behaviours in SSD. Study design: Observational studies, or secondary analysis of clinical trials access to full published text and methodology must be available English language published after 1980 (when bulimia nervosa was added to DSM III)	case-reports or case-series and non-empirical designs; reviews, book reviews, comments, conference proceedings, abstracts only publications, duplicate publications, and grey literature.

**Table 3 nutrients-13-03820-t003:** Summary of included studies.

Author/Year/Country	Disordered Eating Studied and Instrument(s) Used	Objectives/Aims	Methods	Strengths/Limitations	Main Findings
Abbas and Liddle, 2013 UK	Measured food cravings using the food craving inventory (FCI).	To compare carbohydrate and other food cravings in people with schizophrenia versus healthy controls.	Case-control study comparing 40 people with ICD-10 diagnosis of schizophrenia (20 on olanzapine and 20 on typical antipsychotics) and 20 healthy controls	Strengths included clearly defined research question, selection criteria, control group, sample size calculation, use of validated measures, and adjustment for confounding (albeit limited). Limitations include diversity of settings that samples were drawn from, and those on typical antipsychotics were mostly on depot antipsychotics.	Mean age 39.4 (11.7) for cases, and 40.9 (11) years for controls. M: F 29:31. Patients on typical antipsychotics had higher craving scores, but this was not statistically significant.
Aguiar-Bloemer et al. 2018 Brazil	Measured binge eating, food craving, emotional eating, and snacking using semi-structured questionnaire, dietary record and anthropometric scores.	To assess eating behaviours, food practices, and nutritional and metabolic profile of patients with schizophrenia in a tertiary hospital.	Cross-sectional study of 33 patients with DSM IV diagnosis of schizophrenia.	Strengths included using a mixed quantitative and qualitative to study the impact of medications on eating behaviours. Limitations included small sample size, and cross-sectional nature of the study undertaken using non-validated tools.	Mean age of 41.7 (12.6) years and M:F was 24:7. In this group of patients that were predominantly overweight (~71%) and/or had metabolic syndrome (~42%), 16.1% reported sweet craving. Participants also reported emotional eating, and binge eating symptoms such as eating rapidly, eating large quantities of food (especially when unwell), and eating more frequently, and increased snacking.
Bachman et al. 2012 Germany	Measured dietary restraint, disinhibition, and hunger using three factor eating questionnaire (TFEQ).	To evaluate subjective eating behaviour (among other subjective parameters) in a child and adolescent psychiatric population under long-term antipsychotic treatment in a rehabilitative setting.	Cross-sectional study of 74 adolescent patients with ICD-10 diagnosis of schizophrenia (67), schizoaffective disorder (3), and other diagnosis (4) on clozapine (56) or olanzapine (18).	Only study found in child and adolescent population, with clear research question and performed using validated tools. Limitations include small sample size, poor sample description, non-adjustment for confounders, and focus on psychological concepts underlying abnormal eating behaviours rather than on disordered eating.	Mean age was 19.9 (2.3) years and there were more males (49:25). The restraint score was in the disordered eating range and an association was found between BMI and hunger scales among male patients.
Blouin et al. 2008 Canada	Measured dietary restraint, disinhibition, and susceptibility to hunger using TFEQ.	To explore and compare indicators of eating behaviours including dietary restraint, disinhibition, and susceptibility to hunger among 18 male patients with schizophrenia spectrum disorders (SSD) treated with second generation antipsychotics (SGA) to 20 healthy, sedentary and untreated individuals.	Case-control study of 18 male patients with DSM IV schizophrenia (15), schizoaffective disorder (2), and delusional disorder (1), compared with 20 “non-schizophrenic” healthy men.	Strengths include use of multiple validated tools, and adjustment (although limited) for confounding. Limitations include small sample size, no power calculation, limited observations to variable analysed ratio, and focus on underlying cognitive mechanisms rather than on disordered eating.	Mean age of 30.5 (7.9) years for cases and 29.5 (6.7) years for controls, and all participants were men. Patients treated with SGAs were significantly more likely to be obese and have higher cognitive restraint, disinhibition, and susceptibility to hunger triggered by internal and external cues. The higher disinhibition accounts for overconsumption of food in response to variety of stimuli, and loss of control.
Bromel et al. 1998 Germany	Eating behaviours using a semi-structured questionnaire.	To determine serum leptin concentrations prior to and after initiation of clozapine treatment.	Ten-week, prospective, follow-up study of 12 patients with DSM-IV schizophrenia (9) and schizoaffective disorder (3).	Strengths included clear hypothesis, prospective design, and ability to demonstrate changes that mirror clozapine prescribing in real world. Limitations included small sample size, no power calculation, and poor sample description, use of non-validated questionnaires, and not adjusting for confounders.	Mean age was 31 (7.1) years and there were 6 males and 6 females. There was significant increase in serum leptin that corresponded with increases in mean body weight, mean BMI, mean fat mass, and mean lean body mass. Nine patients reported increase in hunger and appetite and food cravings, and two (16.6%) had features of binge eating disorder (BED).
Case et al. 2010. Multi-site clinical trial	Food craving (with FCI) and eating inventory and other non-validated questionnaires such as eating behaviour assessment (EBA) and eating attitude scale (EAS)	To test the hypothesis that changes in appetite might be indicative of patient’s weight gain during treatment for schizophrenia.	Secondary analysis of data from 4 phase IV clinical trials of patients with SSD or BPAD (varying diagnostic criteria) on olanzapine.	Strength includes the large sample size. The authors attempted to compare results of 4 different industry sponsored trials of varying methodology, patient demographics, diagnostic criteria, medication dose, trial duration, poor sample description, and use of invalidated measures.	Mean age ranged from 35.6 (12.2) years in one trial to 43.5 (9.5) years in another one. Men comprised 55.15% of the total sample. There was a significant correlation between increase in appetite for fatty food and weight increase in trial 3, which showed greatest overall weight change. Score increases of several EBA and EAS items indicate that binge eating showed correlation with weight gain.
De Beaurepaire, 2021 France	Binge eating disorder (BED) and bulimia nervosa (BN) were diagnosed using DSM IV criteria. BED was classified as syndromal or sub-syndromal. Night eating (NE) was evaluated by a single question “do you get up at night to eat?”	To investigate the prevalence of eating disorders (BN, BED, and NE) in patients with schizophrenia and schizoaffective disorder chronically treated with antipsychotic monotherapy, and to investigate whether different antipsychotics or classes of antipsychotics are differentially associated with eating disorders.	Cross-sectional study of 156 outpatients with chronic schizophrenia or schizoaffective disorder on antipsychotic monotherapy for less than 8 years.	The strengths include modest sample size, clear research aims, use of standardised questions/criteria for binge eating. The limitations include poor sample description, use of single item to screen for night eating, and not adjusting for confounders.	The mean age was 41.7 years. There were 88 men and 68 women. Prevalence of BN was 0, syndromal BED was 4.4%, and sub-syndromal BED was 18.7%. Prevalence of night eating was 30%. BED spectrum and night eating was more prevalent in the clozapine/olanzapine group, and the differences were significant in women. BED spectrum disorders were more common in the first two years of treatment, but night eating was stable over time.
Fawzi and Fawzi, 2012 Egypt	Disordered eating attitudes were measured using eating attitudes test (EAT) 40	To test the hypothesis that disordered eating attitudes co-occur with schizophrenia in a higher frequency and that disordered eating comorbidity would be associated with more severe schizophrenia psychopathology.	A case-control study of 50 consecutive antipsychotic naïve patients with DSM IV schizophrenia recruited from outpatient clinic, compared to 50, age, and gender matched healthy controls with no current or lifetime diagnosis of psychiatric disorder.	The strengths included sample size calculation, well-described sample and selection criteria, use of valid instruments, and adjustment for confounders. The limitation includes the study design.	The mean age of cases was 29.4 (10.2) years and that of controls was 31.1 (10.8) years. 29/50 (58%) of cases and controls were men. Disordered eating attitudes as measured by EAT 40 were two and half times more in schizophrenia than in controls (30% versus 12%). Presence of disordered eating behaviour in patients is associated with higher total PANSS scores, female gender, lower leisure time physical activity, and higher tea/coffee use.
Garriga et al. 2019 Spain	Food craving was measured using FCI. A semi-quantitative food frequency questionnaire was used to measure food consumption.	To describe the longitudinal evaluation of food craving in a sample of seriously mentally ill patients starting on clozapine.	An 18-week follow-up study of 34 consecutive patients with DSM IV TR diagnosis of schizophrenia (27), schizoaffective disorder (5), and bipolar disorder (2), who were commenced on clozapine.	The strengths include the prospective design, good sample description, use of validated instruments, and adjusting for confounders. Limitations include small sample size, no power calculations (for multiple analysis), and short-term follow-up.	The mean age was 36.8 (12.2) years and there were 21 men (61.8%). On adjusting for BMI, the normal weight group (at baseline) had significantly increased score for complex carbohydrate and protein food cravings, and were related to male gender, older age, tobacco use, recent onset of psychosis and higher plasma levels of nor-clozapine. The overweight/obese group had higher baseline fast food cravings. Only increments in fast food cravings were associated with weight gain on follow-up.
Goluza et al. 2017 Australia	Food addiction was measured using Yale food addiction scale (YFAS)	To examine the prevalence of food addiction and to explore the associations between participant characteristics and food addiction diagnosis.	Cross-sectional study of final sample of 93 outpatients with diagnosis of schizophrenia from tertiary regional hospital in NSW, Australia.	The strengths include use of valid instrument for food addiction. The authors did not describe their sample, or the diagnostic criteria used, did not adjust for confounders, or provide information about common factors associated with disordered eating.	The authors did not provide mean age. Majority of the sample was between 56 and 65 years of age. There were more males (M:F was 61:32). Most patients were on quetiapine, followed by clozapine and olanzapine. In total, 27% of the sample met criteria for food addiction and among those who did not meet the criteria, 77.4% endorsed ≥ 3 symptoms but did not report distress or impairment criteria. The most common food addiction symptoms were persistent desire or repeated attempts to cut down.
Hay and Hall, 1991 New Zealand	DSM III R eating disorders using Pope and Hudson Eating Questionnaire and a semi-structured interview based on DSM III R.	To assess the point prevalence of eating disorders in recently admitted psychiatric inpatients.	Cross-sectional study of final sample of 101 current psychiatric inpatients recruited over a three-month period and assessed for DSM II R eating disorders.	This early study demonstrated the presence of undetected eating disorders among psychiatric inpatients using standardised tools. Limitations include the facts that no separate information provided for schizophrenia or SSD, poor sample description, poor description of analysis, and no evidence of adjustment for confounders.	The mean age was 36 (15.4) years. There were 59 females and 42 males. In total, 18 patients (17%) had current DSM III R eating disorders (8 with BN and 10 with EDNOS). Of the 10 with EDNOS, 7 had bulimic type, and 4 of this group had SSD diagnosis. In total, 28 other patients had a variety of weight or eating concerns.
Khazaal et al. 2006 Switzerland	Binge eating status was assessed by using SCID for DSM IV. Participants were classified as no bingeing, binge episodes less than 2 days a week (BS), BED, and BN.	To assess binge eating symptomatology in schizophrenia patients receiving treatment and compared to a group of non-psychiatric controls.	Case-control study of 40 patients with DSM IV schizophrenia and 40 non-psychiatric controls. Both groups were divided into severely overweight (BMI > 28) and comparison sample (BMI < 28).	The strengths include clear aims, use of valid tools, and having a comparator group. Limitations include the cross-sectional design, small sample size, poor sample description, and non-adjustment for confounders.	The mean age of patient sample was 33.8 (9.1) and that of control population was 35.5 (10.8) years. There were 19 females with schizophrenia and 21 females in the control group. Patients had higher BMI than the control group. 17/40 patients and 10/40 controls had binge eating (BS or BED). Among patients with BMI > 28, 60% had binge eating symptomatology.
Khazaal et al. 2010 Switzerland	Authors used TFEQ and SCID-IV for AN, BN, and BED.	To study the psychometric properties of the revised Mizes anorectic cognitive questionnaire (MAC-R) in people with SSD on treatment.	Cross-sectional study of 125 patients with DSM IV schizophrenia or schizoaffective disorder recruited from outpatient, and day hospitals.	Strengths included the sample size, use of effect sizes to determine power, reasonable sample description, and use of valid instruments. Limitations included study design. The study was not undertaken to explore or describe BED.	Schizophrenia was the predominant diagnosis (93.8%) followed by schizoaffective disorder (6.2%). The mean age was 35.7 (10.9) years and 57.6% were women. In total, 30.5% of the sample had BED. MAC-R dimensions of rigid weight regulation and fear of weight gain correlated with BED and obesity.
Khosravi, 2020 Iran	Disordered eating behaviours (DEB) using EAT 26.	To investigate the biopsychosocial factors in DEB among schizophrenia patients.	Authors used a case-control design to compare 154 DSM 5 patients with schizophrenia (83 in active phase and 71 in remission) recruited by convenient sampling from hospital and 154 healthy controls from the same area.	Strengths included decent sample size, sample size calculation, clear selection criteria, and use of valid measures. EAT 26 is not a diagnostic instrument and therefore cannot describe the types of DEBs.	The authors did not provide mean age. Majority of those in acute phase were between 40 and 49 years and 20 and 29 years and majority of those in remission group were between 20 and 29 years and 30 and 39 years. Males formed 41.4% of cases and 54.5% of controls. DEBs (defined as score ≥ 20 on EAT 26) were seen in 41.5% of schizophrenia patients and 10.3% of the controls. The biopsychosocial associated with DEBs on multiple linear regression analysis included duration of psychosis, atypical antipsychotic prescription, high levels of anxiety and depression, severity of psychosis (higher PANSS scores), tobacco smoking, and type 2 diabetes mellitus.
Kluge et al. 2007 Germany	Abnormal eating behaviours were studied using a standardised binary scale capturing the presence or absence	To describe the efficacy and tolerability data from a double-blind comparison of clozapine and olanzapine, focusing on abnormal eating behaviour.	Authors recruited 30 patients with DSM IV diagnosis of schizophrenia (26), schizoaffective disorder (3), and schizophreniform (1) disorder who were enrolled in a randomised double-blind controlled trial comparing clozapine and olanzapine. Participants were followed up for 6 weeks.	The strength includes the design. Limitations include poor sample description, small sample size and no power calculation, not providing adequate information on dropouts, and limited information on the scales used.	The mean age of the clozapine group was 36.7 (13) years and for the olanzapine group was 32.8 (8.3) years. There were 18 women and 12 men. The number of patients reporting food cravings and binge eating increased significantly over time. In total, 67% of males and 61% of females reported food cravings and 33% males and 28% females reported binge eating. There were no statistically significant differences between the two antipsychotics on tendency to report food cravings or binge eating. The olanzapine group had numerically more people reporting binge eating, food craving, and weight gain.
Knolle-Veentjer et al., 2008 Germany	Appetite and eating behaviours were studied using FEV German Version of TFEQ and executive function using Behavioural Assessment of the dysexecutive syndrome (BADS).	To study the impact of distinct neuropsychological functions in eating behaviours in schizophrenia.	Case-control study involving 29 stable patients with DSM IV diagnosis of schizophrenia and 23 age, gender, and educational criteria matched healthy subjects. They developed a special board game paradigm for delay in gratification and also studied appetite, and eating behaviours using FEV German Version of TFEQ and executive function using Behavioural Assessment of the dysexecutive syndrome (BADS).	The strengths include clear aims, description of sample, use of validated measures, and authors developing their own board paradigm. Limitations include small sample size, not adjusting for confounders, and unknown effects of negative symptoms on results. Authors focused on cognitive mechanisms rather than disordered eating.	The mean age was 34 (10.81) years for cases and 32 (10.78) controls. There were more males in both the cases (19:10) and controls (17:6). Delay of gratification is significantly correlated with overall executive functioning and is impaired in patients compared with controls. Perceived appetite was significantly negatively correlated with delayed gratification. Higher score on dietary restraint and disinhibition was significantly negatively correlated with executive functions. BMI and restraint were positively correlated.
Kouidrat et al. 2018 France	Measured three factors of restraint, emotional eating, and uncontrolled eating with TFEQ-R 21.	To assess and compare eating behaviours, clinical, and biological data of a sample of schizophrenia patients with healthy controls.	Authors designed a case-control study involving 66 consecutive DSM IV outpatients with schizophrenia or schizoaffective disorder and compared them with 81 healthy controls without any psychiatric history or significant medical illness.	Strengths included clear aims, well-described sample, use of valid measures, and adjusting for potential confounders. Limitations included study design, small sample size, and non-description of type of disordered eating.	In comparison to the controls, the patients were statistically older (44 ± 11 years versus 32 ± 14 years), had more males, were more likely to be smokers, and have higher BMI (30.3 ± 8.2 versus 24 ± 3.3). Almost all patients were on antipsychotics. Schizophrenia patients had higher TFEQ scores on all three factors, which remained significant after adjusting for sex, age, BMI, and smoking status. In the control group, women had significantly higher scores for cognitive restraint and emotional eating, whereas among patients, men had significantly higher scores for emotional eating. Cognitive restraint was significantly higher for men with BMI < 25. No correlation was found for medication use or duration of psychosis.
Kucukerdonmez et al. 2019 Turkey	Food addiction was measured using YFAS.	To study the prevalence of food addiction among schizophrenia patients and to assess whether there is a difference between individuals with and without food addiction in terms of nutritional status and anthropometric assessments.	Cross-sectional study of 104 DSM V schizophrenia patients recruited from the local hospital.	Strengths included decent sample size, use of validated instrument, and clear research aims. Limitations included poor sample description, and no adjustment for confounders.	The average age was 39.4 (10.78) and majority were females (60.8%). In total, 67 patients were on monotherapy and 37 were on more than 2 antipsychotics. Food addiction was found in 60.6% of the sample and more in females (62.9% of females versus 57.1% of men), although this was not statistically significant. In total, 41.3% of those with food addiction were obese and 33.3% were overweight and patients who had food addiction had significantly higher dietary energy intake.
Kurpad et al. 2010 India	An eating behaviour questionnaire that used items of DSM IV BED and other obesogenic behaviours. They defined obesogenic behaviours as a spectrum of eating behaviours that could include inability to leave food behind on the plate, anger when people commented on their eating, cravings for food, night eating, buying snacks, and overeating.	To study the prevalence of BED among patients on treatment for psychosis and to assess the spectrum of eating behaviours and its implications on BMI.	Cross-sectional study of 73 outpatients with ICD-10 diagnosis of schizophrenia or Psychosis NOS.	Strengths included clear aims, and reasonable description of the sample studied. Limitations included small sample size, not using validated instruments, and not adjusting for confounders. Authors did not describe what cravings were present and/or the criteria used for night eating.	In total, 51/73 (69.8%) were below 40 years of age and 38/73 (52.1%) were men. The median duration of antipsychotic use was 3 years, and 51/73 (69.8%) had BMI ≥23 (defined for Indian standards as overweight). None of the patients fulfilled criteria for BED. In total, 9/73 (12%) had binge eating. Binge eating was associated with antidepressant use. In total, 18/73 (25%) reported food cravings and night eating in 4% of the sample. Participants reported eating rapidly (48%), inability to leave food on the plate (29%), eating until uncomfortably full (16%), eating when not hungry (15%), feeling guilty after eating (10%), buying snacks (48%) and overeating (44%).
Lundgren et al. 2014 USA	Participants were measured on weight and lifestyle inventory (WALI).	To examine the factors that are self-identified by individuals with schizophrenia as contributing to weight-regulation.	Case-control study comparing 22 obese patients (≥30 kg/m^2^) with DSM IV schizophrenia (18) or schizoaffective (4) disorder, and 27 obese individuals without history of serious mental illness.	The strengths included clear aims and use of valid tools. Limitations included poor description of sample, small sample size, use of different versions of WALI used by patients and controls, post hoc nature of comparisons, and unknown influence of mood symptoms or socio-demographic variables on disordered eating.	The mean age of cases was 46 (10.2) years and controls was 46.1 (10.8) years. Majority were females (54.5% of cases and 55.6% of controls). Cases were significantly less educated, less likely to be married, more likely to live alone, more likely to be current smokers, report mood problems in the past month and report lifetime physical abuse. In total, 6/22 (27.2%) reported evening hyperphagia and 11/22 (50%) reported getting up in the middle of the night to snack. Patients also reported significantly higher scores for eating in response to sight, smell, and taste, and continuing to eat because they do not feel full after a meal, eating because of physical hunger, eating when alone, and overeating at lunch.
Lyketsos et al. 1985 Greece	DSM III criteria for eating disorders, and schizophrenia eating disorder questionnaire (which covered items of thought, perception, deviant behaviour, eating dysfunctions, and neurotic symptoms), and EAT.	To investigate eating disorders and eating attitudes in a population of chronic schizophrenia patients and to compare them with psychotic affective disorder patients and normal controls.	Case-control study of 137 inpatients with DSM III diagnosis of chronic schizophrenia compared to 22 patients with chronic affective psychosis and 60 normal volunteers.	Strengths include the sample size, and use of valid instruments. Limitations include use of a non-validated instrument (schizophrenia eating disorder questionnaire), poor sample description, and lack of adjustment for disorders. However, this study was completed in 1985.	The median age was 49.56 (range from 21–65 years). The study had 58 men and 79 female patients with schizophrenia. Schizophrenia women were found to fulfil criteria for anorexia nervosa (AN), and bulimia nervosa (BN), engage in binge eating, and be overweight. 23 of 25 women who were bulimic symptoms were overweight. There was no loss of control over binge eating. Patients with schizophrenia reported delusions and hallucinations related to eating.
Malaspina et al. 2019 USA	DSM IV criteria for eating disorders: AN, BN, EDNOS	Post hoc analysis of data on groups of schizophrenia patients with and without premorbid ED. Did not describe a clear aim.	Cross-sectional study of 288 sequential inpatients with DSM IV TR schizophrenia and schizoaffective disorder to study premorbid ED symptoms. They compared premorbid ED in those on treatment versus not on treatment.	The strengths include the sample size, definitions and variables used, and the analyses performed. Limitations include poor sample description, poor aims, lack of clarity whether data were prospective or cross-sectional, and whether the differences reported were between the groups or within the same group. The study was conducted over a long period (1994–2008) and the understanding and description of eating disorders changed over that time.	There were more men than women (182:106) and the mean age of the group was 32.72 (9.32) years. Women were higher among both the groups (with and without premorbid ED). In total, 27/288 met the criteria for ED, most commonly AN, followed by BN, and EDNOS. The group with premorbid ED was significantly more psychotic (gustatory hallucinations), and had disorganisation (unusual thought content) symptoms, both during medication free and treatment phases, and higher depression scores in the treatment phase. Cases with ED had significantly higher IQ scores.
Palmese et al. 2013 USA	Night eating questionnaire and night eating interview.	To examine the frequency and clinical correlates of night eating syndrome (NES) in a sample of obese patients with schizophrenia.	Cross-sectional study of 100 obese/overweight patients with DSM IV TR schizophrenia or schizoaffective disorder.	The strengths included decent sample size, use of validated tools, and adjustment for confounders. Limitations include the cross-sectional design and the less than adequate description of selection criteria.	The mean age was 46.5 (10) years and there were more females (61). The mean BMI was 38.2 (7.7) and majority were African Americans (49). In total, 12% of the sample met full criteria and an additional 10% met partial criteria for NES based on the interview, while the rates with the questionnaire was 8% for full criteria and further 8% for partial criteria. A total of 32% reported little or no control over night eating and 40% reported strong urges to eat. Night eating was associated with insomnia and depression but not current psychotic symptoms, antipsychotic medication use, or substance use disorder.
Ramacciotti et al. 2004 Italy	SCID for DSM IV disorders and eating disorders inventory (EDI) were used.	To describe the frequency of eating disorders in a group of schizophrenia patients.	Authors describe a cross-sectional analysis of 31 outpatients with DSM IV schizophrenia and explored for BED and BN, non-purging type.	Strengths included use of valid tools and clear definition of variables studied. Limitations include small sample size, poor sample description, and not adjusting for confounders.	The mean age was 34.8 (9.2) years for men and 41 (10.1) years for women. There were 25 males and 6 females. In total, 71% were overweight and 62% were obese and all patients were on atypical antipsychotics. Five obese patients (16% of the sample) reported BED and BN non-purging type; all were males. BED group was significantly younger, more obese (non-significant), and scored positively for less drive for thinness on EDI.
Ryu et al. 2013 South Korea	Food craving was measured by general food craving questionnaire trait (GFCQT). Authors also used a locally developed and validated instrument called drug related eating behaviour questionnaire (DREBQ).	To investigate the extent and nature of SGA’s effects on appetite and nature of eating behaviour of schizophrenia patients and to investigate the association between the degree of eating behaviour changes and weight gain during the early phase of antipsychotic treatment.	Prospective study (12 weeks) of 45 patients with DSM IV schizophrenia who were on SGA monotherapy and after a 4-week washout period.	Despite being a brief article, the authors described clear aims, tools that were specifically developed (to capture medication induced eating behaviour changes) and validated, clear description of sample, statistical techniques used to adjust for missing scores, and confounders and the prospective design. Limitations include small sample size, uncertain generalizability (as the tools are not validated in other areas), and short follow-up time.	The mean age was 32.1 (range for 18–50 years) and the male: female ratio was 1:1. Majority of the patients were on Risperidone (24), followed by olanzapine (13). A total of 33% of the sample reported changes in hunger and cravings for sweets. BMI changed over time and overall BMI changes over 12 weeks was significantly associated with total DREBQ score, baseline BMI, and age. DREBQ total score correlated with preoccupation with food and loss of control factors.
Sentissi et al. 2009 France	TFEQ and Dutch eating behaviour questionnaire (DEBQ)	The aim of the study was to gain insight into the effects of different categories of antipsychotic drugs on the food attitudes of schizophrenia patients.	Cross-sectional study of 153 patients with DSM IV schizophrenia recruited from inpatient and outpatient settings.	The strengths include the sample description, clear selection criteria, use of valid tools, and adjustment for confounders. Limitations include small numbers within individual groups making it hard to compare and use self-report measures.	The mean age was 33.1 (8.7) years and there were 94 men (61.4%). The sample included 33 patients who were antipsychotic naïve. The mean BMI was 25.6 (5.5); 23.5% were overweight and 22.9% were obese. In total, 19% had metabolic syndrome and 37.3% had high waist circumference. Women had higher TFEQ restraint and disinhibition scores than men. Patients on atypical antipsychotics were more sensitive to external eating cues and have a greater tendency towards disinhibition scores. Overweight and obese patients have a higher susceptibility to hunger and disinhibition. DEBQ external eating score negatively correlated to total PANSS score and DEBQ emotional and external eating factors were higher in women compared to men. External DEBQ factor was also higher in atypical antipsychotic group compared to those on conventional antipsychotics and antipsychotic naïve group.
Srebnik et al. 2003 USA	EAT-26 was used to study eating disorders. A cut-off score of ≥20 was used to indicate presence of eating disorder.	This pilot study aimed to describe the prevalence of eating disorder symptoms and the clinical and demographic predictors of those symptoms among adults with severe and persisting mental illness (SPMI) receiving community mental health services.	Cross-sectional study of 149 community mental health participants with SPMI, 38% of who had a diagnosis of schizophrenia spectrum diagnosis.	The strengths include clear aims, use of valid tools, and adjustment for confounders. The limitations include poor sample description, small sample size for individual illnesses, and lack of clarity on whether there were any diagnosis specific predictors for eating disorders.	The mean age was 40.8 (9.7) years and 51% of the sample were women. In total, 13% of SSD patients had an EAT score of >20. The mean scores and proportions were lower than in bipolar disorder (33.3%) and depression (27.5%). Purging was more common in SSD, but this was not statistically significant. Female gender and BMI were significant predictors for eating behaviour for the entire sample.
Stauffer et al. 2009 USA	Eating behaviours were studied using eating inventory (EI), and two non-validated tools, the EBA and the visual analog scale (VAS).	To investigate patients’ characteristics and changes in their eating behaviours during treatment with olanzapine and weight mitigating agents in overweight patients. Authors hypothesised that cognitive restraint and changes in eating behaviours may be indicators of subsequent weight gain or weight loss.	Authors undertook post hoc analysis of 16-week RCT data from three industry-sponsored trials in adult patients with DSM IV TR diagnosis of schizophrenia, schizoaffective disorder, schizophreniform disorder, or bipolar disorder and evaluated the efficacy of nizatidine, amantadine, and sibutramine on weight change.	The advantages include prospective data from RCT. The limitations include poor sample description, differences between the three trials (in age, gender, ethnicity, baseline BMI, and eating behaviour measures), and use of non-validated tools. There is no information on the effects of psychotic or mood symptoms on eating scores. The study only included overweight or obese individuals, which limit generalisability.	There were 158 participants overall; the mean ages and gender distribution differed across the three trials. Higher BMI and less interest in food at baseline, decrease in appetite, carbohydrate craving, or hunger over time predicted weight loss. Patients who experienced a decrease in cognitive restraint and increase in hunger or overeating were more likely to gain weight.
Striegel-Moore et al. 1999 USA	Used ICD 9 criteria to diagnose AN, BN, and EDNOS.	Describe eating disorder (AN, BN, and EDNOS) comorbidity in men admitted to VA centres in USA.	Cross-sectional study of 466,590 men admitted to 155 VA centres in the USA.	Despite the large sample size, the study does not provide details of how many patients had a diagnosis of schizophrenia or SSD. The study does not have clear aims, did not describe the sample, or the analysis made, and have not adjusted for confounders.	The study does not provide age and was performed only on males. Of 466,590 men, 98 cases had an eating disorder: 25 had anorexia, 17 with BN, and 56 with EDNOS. Schizophrenia was a comorbid diagnosis in 36% of patients with AN, 18% of patients with BN, and 27% of patients with EDNOS.
Teh et al. 2020 Singapore	EAT-26 score of ≥20 was used to identify disordered eating	To explore potential moderating effects of depression and anxiety levels on relationship between body image disturbances and disordered eating among participants with mental illness.	Cross-sectional study of 329 outpatients with DSM IV diagnosis of SSD, depressive, and substance use disorders.	The strengths included the clear selection criteria, tools used, definitions used, and statistical techniques including adjustment for confounders. There is no separate information available for patients with SSD, and besides a generic description of disordered eating, there is no information on types or symptoms of disordered eating.	SSD formed 47% of the sample. The mean age for this group was 29.6 (5.6) years, and the mean BMI was 26.7 (6), which were greater than that for depressive and substance use disorders. In total, 51.7% of the entire sample were men; the information for SSD is not known. SSD patients scored significantly lower on depression, and anxiety scores and EAT scores. Participants with disordered eating in the entire sample had greater anxiety and depressive scores than those without disordered eating.
Treuer et al. 2009 Multi-site	Questionnaire to explore appetite, hunger, and eating behaviours with items that covered binge eating.	To explore which disease behavioural and lifestyle factors were associated with weight gain in patients switching or initiating treatment with olanzapine for schizophrenia or bipolar mania.	A multi-site prospective, non-interventional, industry sponsored study that recruited individuals with DSM IV TR or ICD-10 diagnosis of bipolar (93), or schizophrenia (527).	The study was well reported with sample size calculation, good sample description, multi-site recruitment, prospective 6-month follow-up study design, and adjusted for confounders. The limitations included use of self-report and non-validated questionnaire and including patients who participated in a weight control program, which limits generalisability.	A total of 622 participants were recruited from 37 sites across China, Taiwan, Romania, and Mexico. The mean age was 32.6 years and 56% were females and the mean BMI was 23.2. Bipolar group had significantly greater mean age, weight, BMI, more females, more Caucasians, and were on more concomitant medications than the schizophrenia group. After 6 months, the mean weight change was 4.1 KG and 43.9% of patients had clinically significant weight gain. Weight gain was associated with meal frequency, evening snack consumption, eating until uncomfortably full, needing an excessive amount of food to feel full and preoccupation with food.

Key for abbreviations used: AN: anorexia nervosa, BMI: body mass index, BPAD: bipolar affective disorder, BED: binge eating disorder, BN: bulimia nervosa, DEB: disordered eating Behaviours, DEBQ: Dutch eating behaviours questionnaire, DREBQ: drug related eating behaviour questionnaire, DSM: diagnostic and statistical manual of mental disorders, EAS: eating attitude scale, EAT: eating attitudes test, EBA: eating behaviour assessment, EDI: eating disorders inventory, EI: eating inventory, EDNOS: eating disorder not otherwise specified, FCI: food craving inventory, GFCQT: general food craving questionnaire trait, ICD: international classification of diseases, MAC-R: Mizes anorectic cognitive questionnaire, NE: night eating, RCT: randomised control trial, SCID: structured clinical interview for DSM, SGA: second generation antipsychotics, SPMI: severe and persisting mental illness, SSD: schizophrenia spectrum disorders, TFEQ: three factor eating questionnaire, VAS: visual analog scale, WALI: weight and lifestyle inventory, YFAS: Yale food addiction.

**Table 4 nutrients-13-03820-t004:** Summary of Quality Assessment.

		Selection Bias			Detection Bias	Attrition Bias	Confounding Bias	
Author & Year	Study Design	Research Question, Aim, or Hypothesis	Sample Size or Power Calculation	Study Population Clearly Defined	Outcome Measure(s) Clearly Defined Used Valid Tools	Dropouts & Statistics	Confounders	Study score & Rating Good: 7–9 Fair: 4–6 Poor: 1–3
Abbas and Liddle, 2013	Case-control	+	+	+	+	+	+	7 Good
Aguiar-Bloemer et al. 2018	Cross-sectional	+	-	+	-	-	-	3 Poor
Bachman et al. 2012	Cross-sectional	+	-	-	+	+	-	4 Fair
Blouin et al. 2008	Case-control	+	-	+	+	+	+	6 Fair
Bromel et al. 1998	Prospective	+	-	-	-	-	-	3 Poor
Case et al. 2010	Post-hoc analysis of 4 trials	+	-	-	-	-	-	3 Poor
De Beaurepaire, 2021	Cross-sectional	+	-	-	+	+	-	4 Fair
Fawzi and Fawzi, 2012	Case-control	+	+	+	+	+	+	7 Good
Garriga et al. 2019	Cohort	+	-	+	+	+	+	7 Good
Goluza et al. 2017	Cross-sectional	+	-	-	+	+	-	4 Fair
Hay and Hall, 1991	Cross-sectional	+	-	-	+	-	-	3 Poor
Khazaal et al. 2006	Case-control	+	-	-	+	+	-	4 Fair
Khazaal et al. 2010	Cross-sectional	+	+	+	+	+	+	7 Good
Khosravi, 2020	Case-control	+	+	+	+	+	+	7 Good
Kluge et al. 2007	Data from RCTs	+	-	-	-	+	-	5 Fair
Knolle-Veentjer et al. 2008	Case-control	+	-	+	+	+	+	6 Fair
Kouidrat et al. 2018	Case-control	+	-	+	+	+	+	6 Fair
Kucukerdonmez et al. 2019	Cross-sectional	+	-	-	+	+	-	4 Fair
Kurpad et al. 2010	Cross-sectional	+	-	+	-	+	-	4 Fair
Lundgren et al. 2014	Case-control	+	-	-	+	+	-	4 Fair
Lyketsos et al. 1985	Case-control	+	-	-	+	+	-	4 Fair
Malaspina et al. 2019	Cross-sectional	-	-	-	+	+	+	4 Fair
Palmese et al. 2013	Cross-sectional	+	-	-	+	+	+	5 Fair
Ramacciotti et al. 2004	Cross-sectional	+	-	-	+	-	-	3 Poor
Ryu et al. 2013	Prospective	+	-	+	+	+	+	7 Good
Sentissi et al. 2009	Cross-sectional	+	-	+	+	+	+	6 Fair
Srebnik et al. 2003	Cross-sectional	+	-	-	+	+	-	4 Fair
Stauffer et al. 2009	RCT data	+	-	-	-	+	+	6 Fair
Striegel-Moore et al. 1999	Cross-sectional	-	-	-	+	-	-	2 Poor
Teh et al. 2020	Cross-sectional	+	+	+	+	+	+	7 Good
Treuer et al. 2009	Prospective	+	+	+	-	+	+	7 Good

## Data Availability

Not applicable.

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
