# Peer review of "Disordered Eating among People with Schizophrenia Spectrum Disorders: A Systematic Review"

_nutrients, 2021, doi:10.3390/nu13113820_

Round 1

Reviewer 1 Report

This is a very interesting systematic review of 31 studies from January 1884 to February 2021. The authors aimed to review literature on disordered eating among people with schizophrenia spectrum disorders. Therefore, they searched studies in English language in seven databases. However, I have some concerns and I also consider the manuscript should be revised and improved.

Abstract

Is Ok, authors provided all information required. However, I am not completely sure if food cravings and food addiction should be reduced to weight gain or food intake. (See Results section)

Introduction.

Authors introduce the topic, stablish the context, and found the gap in the literature, which mean issues in previous published which were focused on dietary behaviors and eating disorders, but not in disordered eating.

Material and Methods

Searching strategy, study selection and data extraction were appropriate.

Results.

Most of the studies were cross-sectional observational studies (n=15) followed by 9 case-control studies, 4 cohort studies and 3 RCTs.

Twenty-seven studies were performed in patients with a diagnosis of schizophrenia or SSD, and 4 studies included patients with multiple diagnoses. I wonder why these last 4 studies were not excluded or those 5 studies which did not describe the diagnostic criteria used.

Most of the participants in the included studies were receiving antipsychotics, mostly second-generation antipsychotics and specifically, olanzapine and clozapine.

Most studies that reported or described disordered eating, BED and other disordering eating behaviors used validated and standardized instruments. However, authors did not report instrument used to assess food cravings, food addiction or night eating. Weight gain could be explained by food cravings, but not be reduced to the experience of food cravings. Furthermore, some studies report a nonlinear association between food cravings and BMI.

Discussion

This is a well-documented section, especially the role of medication. However, there are many assumptions regarding food cravings, food addiction or night eating which are not fully covered by evidence obtained in the Results section.  Authors should be more careful in this respect and will highlight what instruments were used in referred studies to support the inclusion of these constructs.

Conclusion.

Conclusions could be Ok in a revised and better documented Results and Discussion sections. Please, consider my concerns.

Author Response

Rev 1

This is a very interesting systematic review of 31 studies from January 1884 to February 2021. The authors aimed to review literature on disordered eating among people with schizophrenia spectrum disorders. Therefore, they searched studies in English language in seven databases. However, I have some concerns and I also consider the manuscript should be revised and improved.

Abstract

Is Ok, authors provided all information required. However, I am not completely sure if food cravings and food addiction should be reduced to weight gain or food intake. (See Results section)

Author response: Thank-you for this point; we have now edited the abstract, so this section is clearly articulated. We have been mindful not to exceed the abstract word limit.

Introduction.

Authors introduce the topic, stablish the context, and found the gap in the literature, which mean issues in previous published which were focused on dietary behaviors and eating disorders, but not in disordered eating.

Material and Methods

Searching strategy, study selection and data extraction were appropriate.

Results.

Most of the studies were cross-sectional observational studies (n=15) followed by 9 case-control studies, 4 cohort studies and 3 RCTs.

Twenty-seven studies were performed in patients with a diagnosis of schizophrenia or SSD, and 4 studies included patients with multiple diagnoses. I wonder why these last 4 studies were not excluded or those 5 studies which did not describe the diagnostic criteria used.

Author response: We acknowledge these articles are limited in quality for the said reasons, but they provide important information that contribute to our knowledge on the topic and are summarised in our review. We believe it is better managed by including these studies and describing their findings but also commenting on the limitations.

Most of the participants in the included studies were receiving antipsychotics, mostly second-generation antipsychotics and specifically, olanzapine and clozapine.

Most studies that reported or described disordered eating, BED and other disordering eating behaviors used validated and standardized instruments. However, authors did not report instrument used to assess food cravings, food addiction or night eating.

Author response: Thank-you for this point; we agree it is an important aspect to consider. We have added this to the section under food craving and food addiction, and night eating within the results (page 24).

Weight gain could be explained by food cravings, but not be reduced to the experience of food cravings. Furthermore, some studies report a nonlinear association between food cravings and BMI.

Author response: Thank-you for this point. We have captured this information in our results section. “Craving for fast-food was associated with weight gain [28, 41] and food addiction was associated with increased dietary intake [49]. Stauffer et al [57] demonstrated that reduction in carbohydrate craving was predictive of weight loss in treatment trials”

Discussion

This is a well-documented section, especially the role of medication. However, there are many assumptions regarding food cravings, food addiction or night eating which are not fully covered by evidence obtained in the Results section.  Authors should be more careful in this respect and will highlight what instruments were used in referred studies to support the inclusion of these constructs.

Conclusion.

Conclusions could be Ok in a revised and better documented Results and Discussion sections. Please, consider my concerns.

Author reply: We have addressed these concerns and these are captured in our earlier responses.

Reviewer 2 Report

I appreciate the opportunity to review this interesting work. The methodology, following the PRISMA guide, seems adequate to me, the results well presented and the discussion consistent. I have just a few small suggestions for the authors.

- Formal aspects.
1.- In the title eating, sometimes it is capitalized and sometimes not.
2.- I don't see the need for a difference between disoredered eating and the other two types in the title, when the first one could encompass them. It would only make sense if only those three types were studied clearly and exclusively. The abstract clearly indicates that the objective is to study disordered eating, without differentiating, .. although subtypes are later qualified, even those that are not in the title, because they have been studied.
3.- The data included in "box 1", I suggest that they be put in table format, and be table 1.

- Aspects of content.

- The findings that disordered eating symptomatology is present in SSD in high prevalence, even before the use of antipsychotics, suggests that the explanation that it is at the core of SSD disorders makes a lot of sense. I think the authors could expand this possible theory further, and reflect on whether in the future any symptoms of disordered eating could be included among the diagnostic criteria for SSD disorders.

Author Response

Rev 2:

I appreciate the opportunity to review this interesting work. The methodology, following the PRISMA guide, seems adequate to me, the results well presented and the discussion consistent. I have just a few small suggestions for the authors.

- Formal aspects.

1.- In the title eating, sometimes it is capitalized and sometimes not.

Author reply: We have corrected this inconsistency

2.- I don't see the need for a difference between disoredered eating and the other two types in the title, when the first one could encompass them. It would only make sense if only those three types were studied clearly and exclusively. The abstract clearly indicates that the objective is to study disordered eating, without differentiating, .. although subtypes are later qualified, even those that are not in the title, because they have been studied.

Author reply: We agree with the reviewer and have amended the title

3.- The data included in "box 1", I suggest that they be put in table format, and be table 1.

Author reply: We have put data in box 1 into a new Table 1.

- Aspects of content.

- The findings that disordered eating symptomatology is present in SSD in high prevalence, even before the use of antipsychotics, suggests that the explanation that it is at the core of SSD disorders makes a lot of sense. I think the authors could expand this possible theory further, and reflect on whether in the future any symptoms of disordered eating could be included among the diagnostic criteria for SSD disorders.

Author reply: We agree with the reviewer; we have speculated on the possible association between comorbid diabetes and disordered eating as a way to explain disordered eating in SSD. We have also added the sentence that disordered eating should be included among diagnostic criteria for schizophrenia.

Round 2

Reviewer 1 Report

Ok
